# Mutation-specific reporter for optimization and enrichment of prime editing

I. F. Schene [1,2,3,9], I. P. Joore[1,2,3,9], J. H. L. Baijens [4], R. Stevelink[5], G. Kok [1,2,3], S. Shehata[1,2,3], E. F. Ilcken[2,3], E. C. M. Nieuwenhuis[2,3], D. P. Bolhuis[2,3], R. C. M. van Rees[2,3], S. A. Spelier[3,6], H. P. J. van der Doef[7], J. M. Beekman[3,6], R. H. J. Houwen[1], E. E. S. Nieuwenhuis[1,8] & S. A. Fuchs [2,3✉]

Prime editing is a versatile genome-editing technique that shows great promise for the generation and repair of patient mutations. However, some genomic sites are difficult to edit and optimal design of prime-editing tools remains elusive. Here we present a fluorescent prime editing and enrichment reporter (fluoPEER), which can be tailored to any genomic target site. This system rapidly and faithfully ranks the efficiency of prime edit guide RNAs (pegRNAs) combined with any prime editor variant. We apply fluoPEER to instruct correction of pathogenic variants in patient cells and find that plasmid editing enriches for genomic editing up to 3-fold compared to conventional enrichment strategies. DNA repair and cell cycle-related genes are enriched in the transcriptome of edited cells. Stalling cells in the G1/S boundary increases prime editing efficiency up to 30%. Together, our results show that fluoPEER can be employed for rapid and efficient correction of patient cells, selection of gene-edited cells, and elucidation of cellular mechanisms needed for successful prime editing.

[1] Division of Pediatric Gastroenterology, Wilhelmina Children's Hospital, University Medical Center Utrecht, Lundlaan 6, 3584 EA Utrecht, The Netherlands. [2] Department of Metabolic Diseases, Wilhelmina Children's Hospital, University Medical Center Utrecht, Lundlaan 6, 3584 EA Utrecht, The Netherlands. [3] Regenerative Medicine Center Utrecht, Uppsalalaan 8, 3584 CT Utrecht, The Netherlands. [4] Utrecht University Graduate School of Life Sciences, Heidelberglaan 8, 3584 CS Utrecht, The Netherlands. [5] Department of Genetics, UMC Utrecht Brain Center, University Medical Center Utrecht, Utrecht University, Heidelberglaan 100, 3584 CX Utrecht, The Netherlands. [6] Department of Pediatric Respiratory Medicine, Wilhelmina Children's Hospital, University Medical Center, Utrecht University, 3584 EA Utrecht, The Netherlands. [7] Department of Pediatric Gastroenterology, University Medical Center Groningen, Hanzeplein 1, 9713 GZ Groningen, The Netherlands. [8] Department of Sciences, University College Roosevelt, Lange Noordstraat 1, 4331 CB Middelburg, The Netherlands. [9] These authors contributed equally: I. F. Schene, I. P. Joore. ✉email: S.Fuchs@umcutrecht.nl

Prime editing is a precise genome-editing technique able to generate any moderately sized genetic variation without the need for double-strand DNA breaks[1]. This technique allows repair of patient mutations, generation of disease models, and in vivo editing[2,3]. The prime editor protein consists of a modified Cas9 protein creating single-strand DNA breaks (nicking Cas9) fused to a reverse transcriptase. This fusion protein forms a complex with a prime editing guide RNA (pegRNA). The pegRNA consists of a spacer defining the target site, a single-guide RNA (sgRNA) scaffold, and an extension encoding the intended edit (Fig. 1a). More specifically, the spacer region of the pegRNA is homologous to the genomic target region and guides the prime editor to create a single-strand DNA break (nick) at the specified location. The pegRNA extension consists of a primer binding site (PBS) and a reverse transcriptase template (RTT). The PBS is homologous to the target region and binds to the nicked DNA strand, priming the reverse transcriptase to elongate the nicked DNA strand based on the sequence encoded by the RTT. The elongation of the nicked strand forms a 3′ DNA flap containing the intended edit. Finally, the 3′ flap is resolved by cellular DNA repair pathways, resulting in genomic incorporation of the intended edit (Fig. 1a).

The technology of prime editing undergoes rapid optimization, including adaptations to the pegRNA to increase degradation resistance (epegRNAs[4]), improvements to the prime editor to increase editing efficiency[5] and to increase the flexibility to allow editing in different genomic contexts[6]. Specifically, the increase in flexibility of prime editors has improved employability beyond the original prime editor (PE2), which requires a protospacer adjacent 'NGG' motif (NGG-PAM) to edit DNA. On average, the pathogenic mutations described in the ClinVar database have only one nearby NGG-PAM (Supplementary Fig. 1). This limits pegRNA design to often suboptimal spacers and PBSs and thereby reduces gene-editing efficiency. Prime editor variants with flexible PAM recognition[6–8] provide at least nine additional PAM sites to target the pathogenic mutations described within the ClinVar database and thereby greatly improve gene-editing potential (Supplementary Fig. 1).

With the rise of new prime editing machinery and the complex mechanisms underlying editing efficiency of (e)pegRNAs, designing optimal prime editing strategies remains largely elusive. A number of pegRNA design tools have been developed, but these are not data-driven and lack extensive validation[9–11]. For the original NGG-PAM-dependent prime editor protein, a deep learning strategy was developed to specify pegRNA characteristics relevant to editing efficiency[12]. Although informative, this deep learning algorithm failed to predict a pegRNA efficiency score for >80% of the mutations in the ClinVar database and does not support non-NGG-pegRNA prediction (Supplementary Fig. 2a, b). As expected, for those pathogenic mutations within the scope of the prediction algorithm, a higher number of available PAMs per mutation resulted in a higher predicted maximum pegRNA efficiency (Supplementary Fig. 2c).

To address the shortcomings of current prediction methods and harness the strength of next-generation prime editing machinery, we present fluoPEER: a fluorescent prime editing and enrichment reporter. Previously developed genome editing reporters for cutting Cas9 and base editors are not applicable to prime editing[13–17]. Furthermore, pegRNAs designed to repair patient mutations can currently only be tested in patient cells. FluoPEER can be tailored to any genomic target site and allows high-throughput analysis of pegRNA designs and prime editors in the cell line of choice using fluorescence-activated cell sorting (FACS) within 7 days from the start of cloning. Moreover, fluoPEER provides a selection method to enrich for genomic editing up to 3-fold relative to conventional selection of transfected cells.

## Results

### FluoPEER guides pegRNA design for efficient genomic editing.
To test the efficiency of various pegRNA and prime editor combinations, we cloned genomic target regions (45–100 nucleotides) into the fluoPEER plasmid. Specifically, the target region was inserted between an (e)GFP and an (m)Cherry cassette under constitutive expression (Fig. 1a). If the genomic target mutation results in a premature stop codon (nonsense) or a frameshift, the unmodified genomic target site can be inserted into the fluoPEER plasmid, resulting in a construct that only expresses GFP. Successful prime editing of the fluoPEER plasmid results in activation of Cherry expression. If the genomic target mutation is not a nonsense or frameshift mutation, either one is added to the target insert in the fluoPEER plasmid (Fig. 1a). We hypothesized that this adaptation would not limit the predictive power of fluoPEER since the substitution or insertion of single nucleotides in the RTT-region has minor effects on prime-editing efficiency (Supplementary Fig. 3). Importantly, this system allows targeting of the reporter and the corresponding genomic locus with the same pegRNA design.

We transfected fluoPEER together with prime editing machinery into HEK293T cells and analyzed reporter editing using FACS after 3 days (Fig. 1b). Based on the ratio of Cherry to GFP signal, an efficiency score was established for each editing condition, enabling ranking of prime editor variants and pegRNAs (Fig. 1c). To verify the reliability of this reporter system, we tested combinations of prime editor variants and pegRNAs to convert a genomically integrated GFP gene to BFP. The same prime editor-pegRNA combinations were used to edit a GFP-derived target region that was inserted into the fluoPEER plasmid (Fig. 1d and Supplementary Fig. 4a). When ranking the editing conditions based on efficiency for both strategies, the correlation between genomic editing and reporter prediction was strong ($R = 0.94$, Fig. 1d and Supplementary Fig. 4b, c). This correlation further confirms that insertion of a 1-nucleotide frame-shift mutation in the reporter does not affect the predictive capacity. Importantly, Sanger sequencing only detected genomic editing in conditions with >10% BFP+ cells (Supplementary Fig. 4c), supporting the need for a sensitive reporter system when targeting difficult-to-edit loci. Next, we tested the ability of fluoPEER to predict the efficiency of prime editor-pegRNA combinations for three different genomic mutations, including a substitution, insertion, and deletion mutation. Again, we found strong correlations between the fluoPEER efficiency score and genomic editing as quantified by next-generation sequencing (Fig. 1e).

We also compared fluoPEER to the DeepPE pegRNA efficiency prediction algorithm, which only supports pegRNA prediction on NGG-PAM sites. Testing combinations of NGG-PE2 with three to four different pegRNAs for six genomic target regions, we found that fluoPEER outperformed the DeepPE algorithm on all targets (Supplementary Fig. 5). Furthermore, we confirmed the robustness of the Cherry over GFP ratio as a read-out for efficiency by varying the input reporter plasmid concentration (Supplementary Fig. 6a, b). By varying the time point of read-out, we further corroborated that the ratio-based ranking was stable over a period of 6 days (Supplementary Fig. 6c). Finally, we confirmed that fluorescent activity disappears from fluoPEER-transfected cells within around 2–3 weeks (Supplementary Fig. 6d).

Using the reporter system, we tested pegRNAs and prime editor variant combinations for various genomic loci. Using available knowledge of characteristics influencing editing efficiency[12,18], we designed pegRNAs containing spacers with high Cas9 single guide RNA (sgRNA) binding scores, PBSs of 9–15 nucleotides length containing at least 5 guanine or cytosine (G/C) nucleotides, and RTTs of 10–20 nucleotides length (Supplementary Fig. 7). Selection of the pegRNA-prime editor condition with the highest reporter prediction score enabled quick and efficient repair of disease-

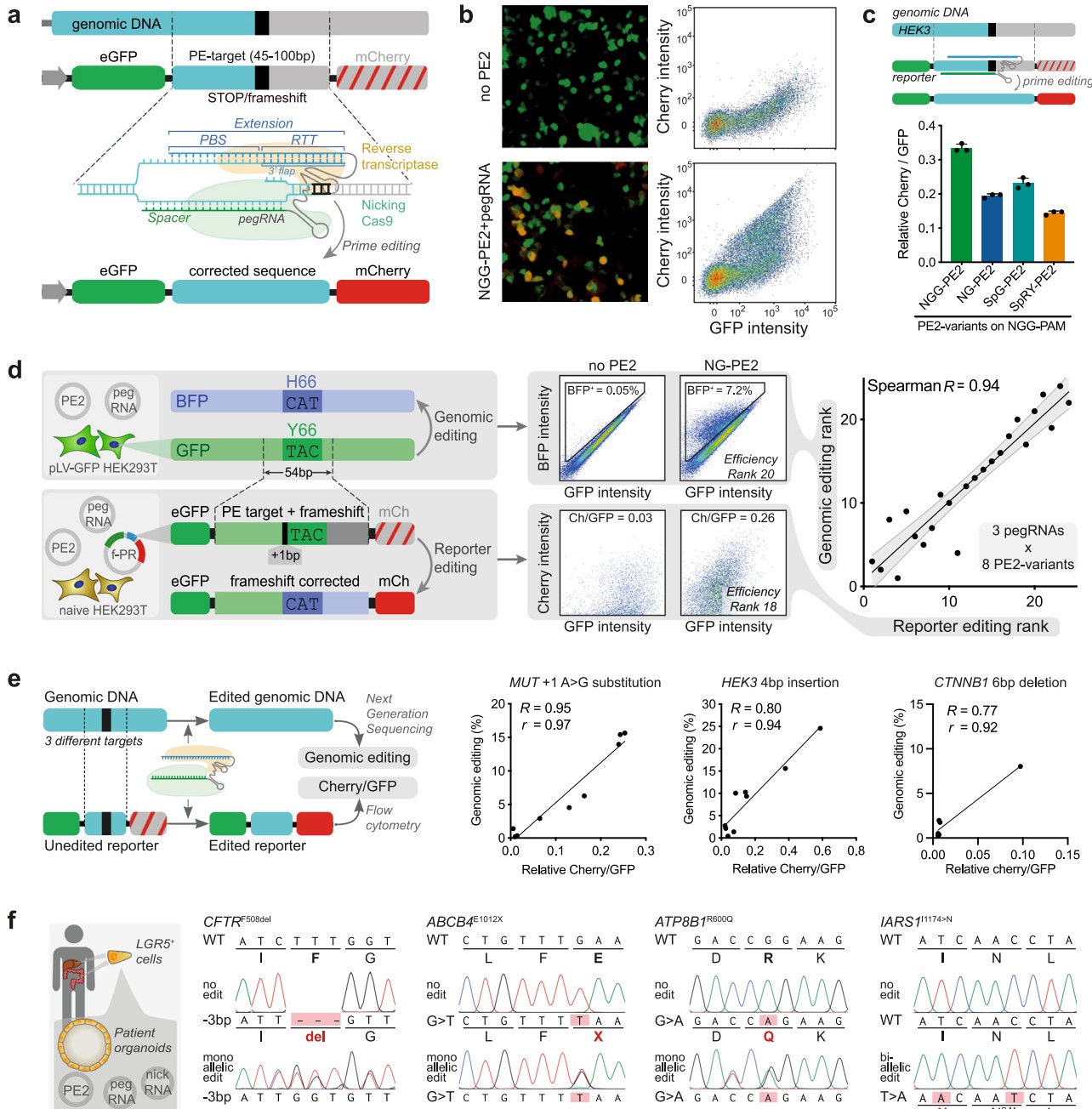

**Fig. 1 FluoPEER instructs pegRNA design for genomic editing. a** The fluoPEER plasmid uses a 45–100 nucleotide genomic region containing a stop codon or frameshift between sequences encoding two fluorescent proteins (eGFP and mCherry). In case the genomic region does not contain a naturally occurring stop codon or frameshift, one is added. The prime edit machinery targets and edits the genomic insert, removing the insertion or stop codon, leading to expression of mCherry in addition to eGFP. The same prime edit machinery, including the same pegRNA design, can edit the genomic DNA. A finished fluoPEER plasmid contains CMV-GFP-P2A-Genomic insert-P2A-Cherry. See method section for more details. **b** Editing of the genomic insert is visualized by Cherry signal and quantified using flow cytometry. **c** FluoPEER distinguishes between the efficiency of different prime editor (PE2) variants. This is quantified as the average ratio of Cherry signal over GFP signal of each transfected HEK293T cell, which gives a measure of editing per transfected plasmid. $n = 3$ biologically independent replicates. **d** Comparison between *GFP* to *BFP* conversion using prime editing on genomic DNA and the fluoPEER plasmid. HEK293T cells containing a lentivirally integrated genomic GFP cassette were transfected with prime editing machinery to convert *GFP* to *BFP* in 24 conditions. The same conditions were applied to HEK293T cells co-transfected with the fluoPEER plasmid containing the sequence encoding the *GFP* to *BFP* conversion. $R$ = Spearman correlation. Gray area represents 95% confidence interval of the linear regression line. **e** For three different genomic targets, various pegRNA designs to install either a substitution, deletion, or insertion mutation were tested in HEK293T cells and editing outcomes were measured using next-generation sequencing (NGS). Corresponding genomic targets were inserted into fluoPEER and the efficiency ratio was extracted. $R$ = Spearman correlation and $r$ = Pearson correlation. **f** Using the optimal pegRNA-prime editor combination based on fluoPEER ranking (Supplementary Fig. 7), several pathogenic mutations in patient-derived organoids were genetically corrected and organoids with biallelic *IARS1* mutations were generated in wildtype liver organoids. Error bars represent standard deviations from the mean. Source data are provided as a Source Data file.

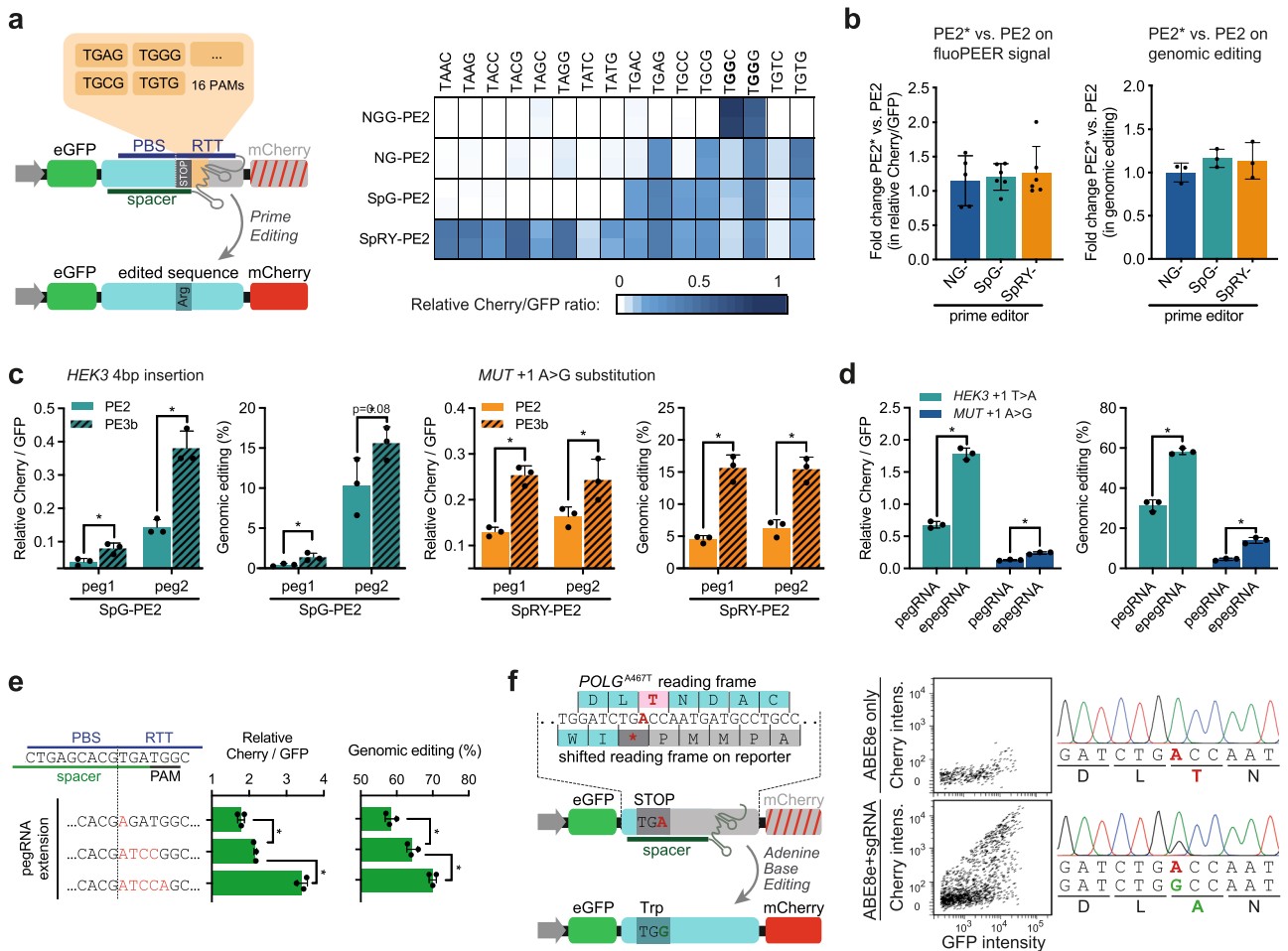

**Fig. 2 FluoPEER enables characterization of various genome editing techniques. a** 16 fluoPEER plasmids were constructed to contain the same pegRNA spacer binding site (*HEK3*) including a stop codon, followed by 16 different 4-nucleotide PAM sequences. Prime editing using a pegRNA that converts this stop codon to an arginine-encoding codon results in Cherry signal. The heatmap shows the fluoPEER signal for these 16 plasmids, four prime editor (PE2) variants, and replicates (*n* = 2) per condition in HEK293T cells. **b** Prime editor variants with adapted nuclear localization sequences (PE2*) were tested in HEK293T cells. Graphs show a summary of all fluoPEER scores (left panel) and of the conversion efficiency of the genomic mutation (right panel), expressed as the fold change (FC) of the PE2* variants relative to the corresponding PE2 variants. See also Supplementary Fig. 8. *n* = 2–3 biologically independent replicates. **c** FluoPEER was used to evaluate nicking sgRNAs for the PE3b technique in HEK293T cells. PE3b designs increased fluoPEER scores (Cherry/GFP ratios) as well as genomic editing as measured by NGS. **d** Improved prime editing using epegRNAs was measured on fluoPEER. Cherry over GFP was measured on fluoPEER; genomic editing was measured using NGS. **e** Increased mismatches between the pegRNA RTT and the target sequence resulted in higher prime editing efficiency on fluoPEER and the genome. Three pegRNAs with varying mismatches with the genome were adopted from Chen et al.[5]. More mismatches resulted in higher editing efficiency on the genome and fluoPEER. Significance was analyzed using a two-tailed unpaired Student's *t* test (**P* < 0.05) for *n* = 3 biologically independent replicates for **c**–**e**. **f** FluoPEER can report base editing when editing of the target nucleotide resolves a stop codon in any possible reading frame. The genomic region of the *POLG*[A467T] mutation was inserted into fluoPEER with a shifted reading frame to create a stop codon (left). This fluoPEER was transfected into fibroblasts with biallelic *POLG*[A467T] mutations to show base editing on the reporter (middle) and the genome (right). Error bars represent standard deviations from the mean. Source data are provided as a Source Data file.

causing mutations in patient-derived organoids (Fig. 1f). These included common mutations causing cystic fibrosis (*CFTR*[F508del] and *CFTR*[G542X]) that are difficult to edit without the use of prime editors with flexible PAM recognition. In addition, we corrected mutations causing progressive familial intrahepatic cholestasis (PFIC) type 3 (*ABCB4*[E1012X]), and PFIC type 1 (*ATP8B1*[R600Q]). Furthermore, we used fluoPEER ranking to efficiently mutate the gene coding for cytosolic isoleucyl-tRNA synthetase (*IARS1*) in liver-derived organoids, generating both biallelic and monoallelic clones (Supplementary Fig. 7b). In these disease models, we confirmed that biallelic, but not monoallelic, *IARS1*[I1174N] mutations impaired organoid expandability, reflecting the failure to thrive observed in patients with biallelic *IARS1* mutations[19,20] (Supplementary Fig. 7b, c). Importantly, pegRNAs targeting a difficult-to-edit locus (low GC-content) to repair a common methylmalonic

acidemia mutation (*MUT*[R369H]) failed to edit the reporter (Supplementary Fig. 7a) and also failed in organoids, demonstrating accurate negative prediction (Supplementary Fig. 7d).

**FluoPEER enables characterization of various genome editing techniques.** To better characterize PE2 protein variants with flexible PAM recognition, we adapted the genomic target region of fluoPEER to test PAM specificities. We found that SpG-PE2 and NG-PE2 have comparable flexible PAM preferences, but that SpG-PE2 has higher efficiency scores overall (Fig. 2a). SpRY-PE2 was most flexible in PAM recognition, essentially functioning as a PAM-less prime editor. Interestingly, a guanine nucleotide on the 4th PAM position led to higher editing efficiency compared to cytosine (Fig. 2a). Overall, the NG-, SpG-, and SpRY-PE2 protein

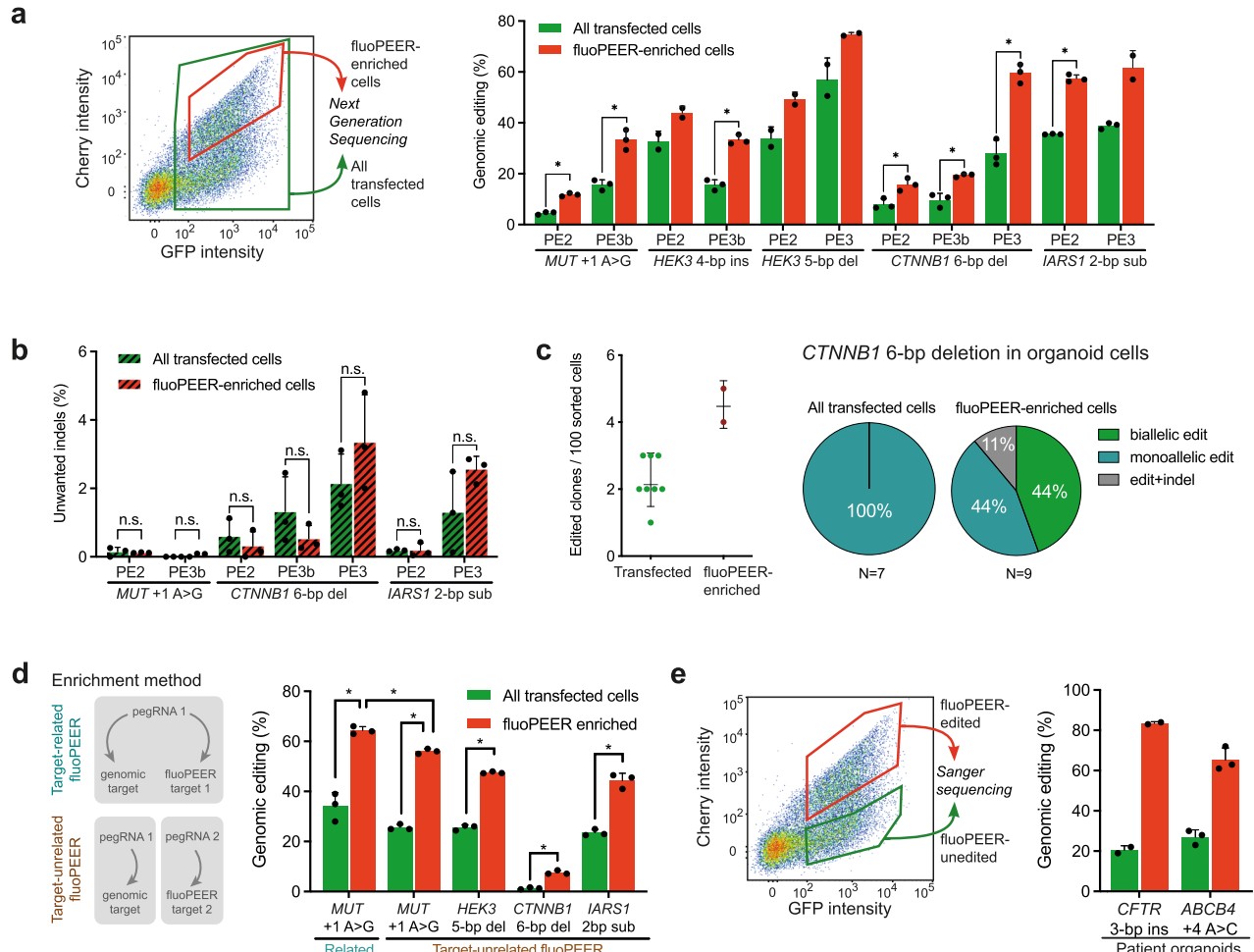

**Fig. 3 FluoPEER enriches for genomic editing. a** FACS sorting based on transfection of the reporter plasmid (GFP+) and presence of reporter editing (GFP+Cherry+) shows enrichment for genomic editing of various genes for PE2 and PE3(b) in the reporter-edited HEK293T cells. Successful editing was quantified by NGS. Note that the *HEK3* 4-bp insertion PE2 condition was performed with NGG-PE2, while the corresponding PE3b condition was performed with SpG-PE2, resulting in lower editing. **b** FluoPEER-enrichment of genomic editing does not increase unwanted indels in HEK293T cells, as quantified by NGS. Significance was analyzed using a two-tailed unpaired Student's *t* test (*P < 0.05) for *n* = 3 biologically independent replicates for **a** and **b**. **c** Activating *CTNNB1* mutations in liver-derived organoid cells allow sustained organoid growth despite removal of Wnt-activator Rspo1 from the culture medium[2]. When creating an activating 6-bp deletion in *CTNNB1* by prime editing, FluoPEER-enrichment resulted in outgrowth of more Rspo1-independent liver organoid clones, compared to regular transfection sorting. From the clones with activating *CTNNB1* mutations, only the clones obtained by fluoPEER-enrichment contained biallelic *CTNNB1* mutations. **d** Use of an unrelated fluoPEER allows enrichment for a genomic edit. Either HEK293T cells were transfected with the fluoPEER corresponding to the genomic mutation or transfected with a fluoPEER unrelated to the genomic mutation. It should be noted that enrichment with the 'related' fluoPEER still yields the highest editing percentage. Significance was analyzed using a two-tailed unpaired Student's *t* test (*P < 0.05) for *n* = 3 biologically independent replicates. **e** Pathogenic mutations in patient colon (*CFTR*^F508del) and liver (*ABCB4*^E1012X) organoids were targeted by PE3 and sorted 72 h after transfection based on fluoPEER editing. Reporter-edited organoid cells were enriched for genomic editing compared to reporter-unedited organoid cells. Error bars represent standard deviations from the mean of *n* = 2–3 biologically independent replicates. Source data are provided as a Source Data file.

variants displayed PAM-specificity patterns highly similar to their corresponding cutting Cas9 proteins[8]. We also tested PE2* protein variants with improved nuclear localization[3], but these variants did not display higher efficiency of reporter editing and genomic editing compared to standard PE2 protein variants in HEK293T cells for the tested mutations (Fig. 2b and Supplementary Fig. 8a, b).

We continued to characterize the ability of the fluoPEER system to report genome editing techniques beyond the standard prime editing strategy of combining a pegRNA and a prime editor. The addition of a nicking sgRNA that only binds on the unedited strand after successful editing of the edited strand is known as PE3b and can increase prime editing efficiency[1]. Indeed, fluoPEER accurately predicted PE3b conditions to

increase genomic editing efficiency (Fig. 2c). Next, we tested the recently reported degradation-resistant epegRNAs[4] and found higher editing efficiency compared to the original pegRNAs, both on the fluoPEER system and genomic DNA (Fig. 2d). Recently, DNA mismatch repair (MMR) was found to inhibit the installment of substitution mutations by prime editing. By increasing the number of mismatches between the RTT and the genome, MMR may be evaded and editing efficiency enhanced[5]. We confirmed this mechanism, which increased editing efficiency both on the reporter and the genome (Fig. 2e). To evade MMR, a new prime editor construct was developed, expressing a dominant negative MMR-disrupting MLH1, called PE4max[5]. Interestingly, we found no significant differences in editing efficiency between PE3 (PE2 + nicking sgRNA) and PE5 (PE4max + nicking

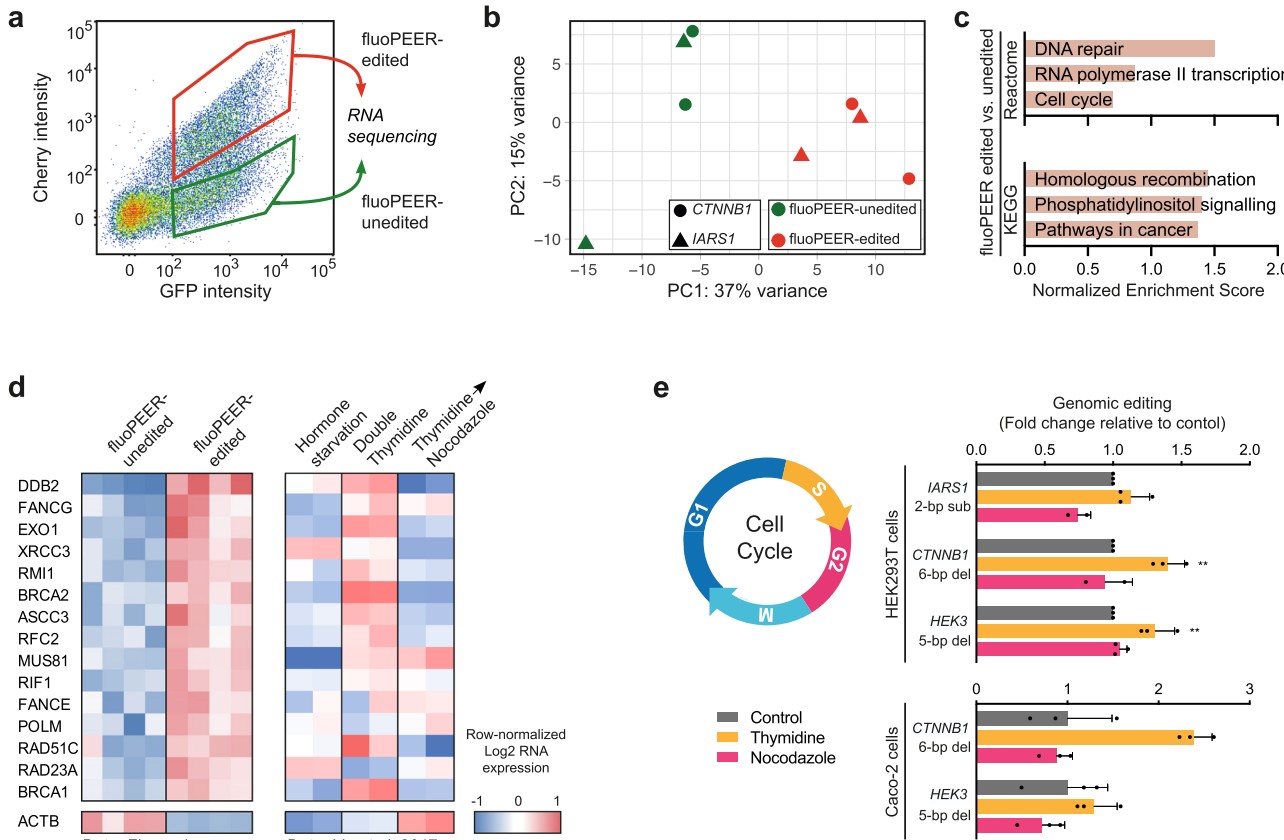

**Fig. 4 Endogenous DNA repair proteins and the cell cycle affect prime editing outcomes. a** Schematic overview of RNA-sequencing set-up. FluoPEER-edited and -unedited HEK293T cells were sorted separately and RNA was sequenced. **b** PCA plot and **c** gene set enrichment analysis of RNA sequencing of the fluoPEER-edited and -unedited cell populations show expression-based differences in DNA repair- and cell cycle-associated genes. **d** Left panel shows a heatmap of the expression of DNA-repair-associated genes which were enriched in the transcriptome of fluoPEER-edited vs. fluoPEER-unedited HEK293T cells from **a–c**. Right panel shows expression of the same genes in publicly available transcriptome profiles (GSE94479) of MCF-7 cells stalled in different cell cycle phases[49]. For both transcriptomic datasets, log2-transformed expression values were mean centered per gene for visualization. **e** Cell cycle synchronization at the G1/S boundary (double thymidine block) or G2/M phase (nocodazole block) affects genomic prime editing efficiency in HEK293T and Caco-2 cells. HEK293T replicates were normalized to the average editing of the control condition for each of three repeated, independent experiments; Caco-2 replicates (n = 3) were normalized to the average editing of the control condition for a single representative experiment with three biological replicates. Error bars represent standard deviations from the mean. Source data are provided as a Source Data file.

sgRNA) for two edits on the genome and fluoPEER in HeLa cells (Supplementary Fig. 8c). This confirms that the effect of suppressing MMR differs greatly between edits[5].

Finally, we tested fluoPEER for base editing, which is possible when base editing resolves a stop codon in any of the six possible reading frames of the genomic target region. We integrated the genomic region of the common POLG[A467T] mutation into the fluoPEER plasmid with a reporter-shifted (+1) reading frame that resulted in a stop codon (Fig. 2f). Adenine base editing (ABE8e-TadA) corrected this mutation on both the plasmid and the genomic DNA of patient-derived fibroblasts (Fig. 2f), showing that fluoPEER could be used for testing base editing gRNA design as well.

**FluoPEER enriches for genomic editing.** The selection of cells with a detectable edit has been shown to enrich for a simultaneously introduced edit at another locus[21,22]. To test whether fluoPEER allows such enrichment, we sorted HEK293T cells that were plasmid-edited (GFP+Cherry+, fluoPEER-enriched population) and found that these cells showed up to 3-fold increases in genomic editing compared to the complete transfected cell population (GFP+) (Fig. 3a). This enrichment of genomic editing did not result in an increase of unwanted insertion and deletion

mutations (indels) in the target region (Fig. 3b). In line with previous findings[1], no mutations could be detected at off-target sites in either transfected or fluoPEER-enriched cell populations (Supplementary Table 1). FluoPEER enrichment of liver-derived organoid cells yielded higher numbers of edited organoid clones. Strikingly, enrichment also enabled generation of clonal organoid lines with biallelic edits (Fig. 3c), which may facilitate in vitro modeling of recessive genetic disorders.

We hypothesized that enrichment of a genomic mutation does not require the use of a 'related' fluoPEER with an insert of the corresponding genomic target region. To test this, we transfected cells with a mix of (1) a pegRNA that targets the genomic DNA, (2) a second pegRNA that targets, (3) an unrelated fluoPEER plasmid, and (4) a prime editor. Indeed, we found that editing of the 'unrelated' fluoPEER plasmid by the second pegRNA enriched for genomic editing by the first pegRNA (Fig. 3d). Importantly, enrichment using a 'related' fluoPEER resulted in slightly higher genomic editing than using a second, unrelated pegRNA-fluoPEER combination (Fig. 3d). Finally, we used the mechanism of fluoPEER enrichment to efficiently generate gene-corrected patient-derived organoids (Fig. 3e). These results establish fluoPEER as a highly dynamic prime-editing read-out, enabling enrichment of genomic editing based on plasmid co-editing.

**Endogenous DNA repair proteins and the cell cycle affect prime editing outcomes.** We considered whether enrichment of genomic editing was caused by higher co-transfection rates of all prime editing plasmids in fluoPEER-edited cells[23]. We therefore transfected a mix of up to four fluorescent plasmids (mTurq2, eGFP, mKO2, mCherry) and tested co-transfection efficiency using FACS. We found that >90% of the cells receiving one fluorescent plasmid also received the three other fluorescent plasmids (Supplementary Fig. 9). This suggests that variance in editing efficiencies within the transfected cell population is not due to unequal co-transfection. In order to investigate differences between reporter-edited and -unedited cells, we compared the transcriptomes of these two populations (Fig. 4a). The transcriptome of reporter-edited cells was enriched for genes associated with DNA repair, specifically homologous recombination, and cell cycle progression (Fig. 4b, c and Supplementary Fig. 10a–d, and Supplementary Data file 1). To further elucidate the association between expression of reporter-enriched DNA repair genes and the cell cycle, we compared our data to publicly available transcriptomes of cells stalled in different cell cycle phases. Interestingly, DNA repair genes upregulated in edited cells were generally higher in cells stalled at the G1/S boundary using a double thymidine treatment (Fig. 4d). We stalled cells at the G1/S and G2/M boundary through application of a double thymidine or nocodazole block, respectively. Directly after releasing cells, we introduced the prime editing machinery and tested genomic editing two days later. For both HEK293T and Caco-2 cells, we observed a general increase in editing efficiency in cells stalled at the G1/S boundary but not in the G2/M phase (Fig. 4e). Using Hoechst staining and fluoPEER, we evaluated prime editing in cycling cells. We found a >40% increase of Cherry signal in cells that were in G2 one day after transfection of prime editing plasmids (Supplementary Fig. 10e). This further confirms an important effect of cell cycle-associated mechanisms, but in this experiment, the precise cell cycle stages that benefit prime editing are difficult to infer due to the delay in Cherry expression. Taken together, successful prime editing was associated with higher expression of DNA repair genes, specifically those expressed at the G1/S boundary. As such, thymidine treatment of cells increased editing efficiency.

## Discussion

We developed fluoPEER as a customizable tool to guide pegRNA design, and to select and optimize prime editor proteins for almost any mutation in patient-derived cells. Furthermore, fluoPEER provides a transient selection method to enrich for genomic editing up to 3-fold compared to conventional selection of transfected cells. Given near-complete co-transfection in our experiments (Supplementary Fig. 9), we used a separate GFP-expressing plasmid and not a prime editor plasmid containing a GFP cassette for this comparison. In contrast to previously developed base-editing reporter and enrichment systems[21,22], this versatile tool is only transiently active, does not require transfection of a second pegRNA, does not rely on genomic integration of a selection cassette, nor requires generation of an additional genomic edit. Moreover, we use fluoPEER to gain insight in the cellular mechanisms underlying prime editing and find cell cycle-related effects on editing efficiency.

Several tools and strategies have been developed to support pegRNA design. DeepPE is a data-driven pegRNA efficiency prediction algorithm, which yielded important insights in general characteristics for pegRNA design[12]. However, the algorithm lacks predictive capacity beyond the NGG-PAM PE2 version of the prime editor. Furthermore, the DeepPE algorithm performs in silico predictions, based on experimentally derived parameters, and can be unreliable in predicting the in vitro efficiency of new

pegRNA designs (Supplementary Fig. 5). An in vitro strategy to evaluate pegRNA designs in high throughput is the use of lentiviral libraries of pooled pegRNAs paired to a DNA target sequence. These pegRNA-target pairs are transduced into cells after which a prime editor is transfected and editing efficiency can be assessed through deep sequencing[12,24]. However, this method only allows testing of pegRNAs with a single prime editor. Conversely, we uncovered optimal combinations by testing pegRNA designs with various (flexible) prime editors.

One proposed method to test combinations of pegRNAs and various prime editors is the Prime Edit Activity Reporter (PEAR). The PEAR depends on the removal of a dysfunctional splice site by prime editing, resulting in activation of GFP expression[25]. It was reported that PEAR can be adjusted to test different pegRNA spacers and PBSs. However, the same splice site-restoring RTT has to be used in all PEAR-targeting pegRNAs, making it impossible to test actual genome-targeting pegRNAs in this system. A more straightforward, but less high-throughput method of testing pegRNA-prime editor combinations is targeting the native genomic site in an easily transfectable cell line, followed by quantification of editing using Sanger sequencing or NGS. However, Sanger sequencing lacks sensitivity, missing sequence variants consisting of up to 15% of total reads (Supplementary Fig. 11). This may lead to underestimation of pegRNA efficiency. Furthermore, although NGS is much more sensitive and reliable than Sanger sequencing in quantifying sequence variants, it is more expensive and time-consuming. Finally, cell lines have wild-type genomes and might therefore be inadequate to test designs that correct pathogenic mutations, such as the $CFTR^{F508del}$ mutation targeted in this work.

Because fluoPEER is more flexible than pooled pegRNA screens, more truthful in pegRNA design than PEAR, more sensitive than Sanger sequencing, and quicker and cheaper than NGS, fluoPEER forms an attractive alternative for optimization of prime editing strategies. Although fluoPEER requires an additional cloning step, this can be performed simultaneously with pegRNA cloning, thereby not requiring additional time. Nevertheless, the preferred method for prime editing optimization might differ between projects and groups, depending on the application and availability of techniques and resources.

The cellular mechanisms underlying successful prime editing are an active topic of research. Our data suggest that the activity of DNA repair mechanisms, specifically homologous recombination, and the cell cycle may influence the outcome of prime editing. A role for the cell cycle has recently been confirmed, with ±1.5 times more efficient prime editing in cycling compared to non-cycling cells[26]. Our results indicate that prime editing is most efficient in cells that are released after stalling in G1/S, which can be linked to the activity of homologous recombination during late S/early G2[27]. Still, while homology directed repair is restricted to dividing cells, prime editing is also active in non-dividing cells[1,26,28]. Furthermore, a large CRISPR interference screen elucidated the effect of 476 DNA repair-related genes on prime editing outcomes[5]. Interestingly, genes involved in homologous recombination were found to mainly prevent indel formation by prime editing, whereas inhibition of mismatch repair-associated genes increased successful editing.

To conclude, fluoPEER is a straightforward and versatile tool that facilitates effective prime editing in various relevant cell types and increases our understanding of cellular processes underlying this genome editing technique.

## Methods

**Study approval and human subjects.** The study was approved by the responsible local ethics committees (Institutional Review Board of the University Medical Center Utrecht and University Medical Center Groningen (STEM: 10-402/K; TcBio 14-008; Metabolic Biobank: 19-489)). For cystic fibrosis organoids, collection of

patient tissue and data was performed following the guidelines of the European Network of Research Ethics Committees (EUREC). Tissue biopsies from the liver of a patient with ABCB4 deficiency (PFIC3) were obtained during a liver transplant procedure in the UMCG, Groningen. Rectal biopsies used for intestinal organoid culture from a patient with ATP8B1 deficiency (PFIC1) and skin biopsies used for fibroblasts culture from a patient with homozygous *POLG*[A467T] mutations were obtained at the outpatient clinic in the UMCU, Utrecht. Biobanked intestinal organoids are stored and cataloged (https://huborganoids.nl/) at the foundation Hubrecht Organoid Technology (http://hub4organoids.eu). All biopsies were used after written informed consent.

**ClinVar database computational analysis**. Information for all pathogenic mutations shorter than 51 bp was obtained from the ClinVar database[29], accessed October 2020. Genomic sequences flanking these mutations were obtained from RefSeq[30] accessed October 2020, using the SPDI data model[31] and a custom python script. The −10 to +4 bp region around the target sites were searched for NGG, NAN, or NGN PAMs. The efficiency of prime editing using NGG PAMs was predicted using PE_Position and PE_type random forest models, provided by Kim et al.[12]. Figures were made in python using Matplotlib[32]. The code used for this analysis is available on GitHub (https://github.com/JBaijens/PE_prediction).

**Plasmid cloning**. FluoPEER plasmids were cloned using the backbone of the pmGFP-P2A-K0-P2A-RFP (Addgene #105686) stalling reporter (SR) plasmid, which was a gift from Ramanujan Hegde. This plasmid was cut directly upstream of the K0-SR domain with SalI and Acc65I for 16 hours at 37 °C, creating 'TCGA' and 'GTAC' overhangs, after which the 6 kb fragment was isolated from gel. Genomic insert oligos containing 5′ 'TCGACC' and 3′ 'G' overhangs on the top oligo, and 5′ 'GTACCC' and 3′ 'GG' overhangs on the bottom oligo were annealed and inserted using a conventional ligation protocol (see Supplementary Note 1). Note that a finalized fluoPEER plasmid still contains the K0-SR domain, which is not shown in the schematic representations in Figs. 1a, d, e and 2a, f as it is not important for the working mechanism of fluoPEER. Cloning of pegRNA and epegRNA plasmids was performed according to previously described protocols[1,4]. In brief, the pU6-pegRNA-GG-Vector (Addgene #132777) or the pU6-tevopreq1-GG-acceptor (Addgene #174038) was digested for 16 h with BsaI-HFv2 (NEB), after which the 2.2 kb fragment was isolated from gel. Oligonucleotide duplexes of the pegRNA spacer, pegRNA extension, and pegRNA scaffold sequences were ordered containing the appropriate overhangs and subsequently annealed. The annealed pegRNA duplexes were ligated into the pU6-pegRNA-GG-Vector using Golden Gate assembly with BsaI-HFv2 (NEB) and T4 DNA ligase (NEB) in a protocol of 12 cycles of 5 min at 16 °C and 5 min at 37 °C. For cloning of sgRNAs used for PE3, we replaced the BsmBI restriction sites of the BPK1520 plasmid with BbsI restriction sites using PCR, which allowed direct ligation of sgRNA-spacer duplexes[33]. All fluoPEER insert, pegRNA, and sgRNA sequences used in this work are listed in Supplementary Data file 2 and were synthesized by Integrated DNA Technologies (IDT). pCMV-PE2 (Addgene #132775), pU6-pegRNA-GG-acceptor (Addgene #132777), and pU6-tevopreq1-GG-acceptor (Addgene #174038) were gifts from David Liu; BPK1520 (Addgene #65777) was a gift from Keith Joung.

**Cloning of flexible PE2s, flexible PE2\*, and SpRY-PE4max**. Using PCR and In-Fusion cloning (Takara Bio), the NGG PAM-recognition domain of the prime editor protein (PE2) was replaced with the corresponding domains in NG-ABEmax[34], SpG-ABEmax, or SpRY-ABEmax[8], to create NG-, SpG-, and SpRY-PE2, respectively. NG-ABEmax was a gift from David Liu (Addgene #124163). SpG- and SpRY-ABEmax were gifts from Benjamin Kleinstiver (Addgene plasmids #140002 and #140003). PE2\* variants with improved nuclear localization sequences (NLSs) were adapted from the NGG-PE2\* developed by Liu et al.[3] and were cloned by PCR and In-Fusion cloning (Takara Bio). Successful cloning of all plasmids was confirmed by Sanger sequencing. Using PCR and In-Fusion cloning (Takara Bio), SpRY-PE4max was created by replacing the PAM-recognition domain of PE4max[5] (#174828) with the corresponding domain of SpRY-ABEmax (#140003)[8]. pCMV-PEmax-P2A-hMLH1dn (PE4max, Addgene #174828) was a gift from David Liu.

**Organoid culture**. Liver and intestinal organoids were grown under standard culture conditions according to previously described protocols[35,36]. In short, liver organoids were plated in matrigel (Corning) and maintained in human liver expansion medium (hL-EM), consisting of AdDMEM/F12 (Gibco) supplemented with, GlutaMAX (1x, Gibco), HEPES (1x, Gibco), PenStrep (1x, Gibco), 2% B27 without vitamin A (Gibco), 1.25 mM *N*-Acetylcysteine (Sigma), 10 mM Nicotinamide (Sigma), 10 nM gastrin (Sigma), 10% RSPO1 conditioned media (homemade), 50 ng/ml EGF (Peprotech), 100 ng/ml FGF10 (Peprotech), 25 ng/ml HGF (Peprotech), 5 mM A83-01 (Tocris), and 10 mM FSK (Tocris). Small intestine and colon organoids were plated in matrigel and maintained in human small intestine expansion medium (hSI-EM), consisting of AdDMEM/F12 (Gibco) supplemented with, GlutaMAX (1x, Gibco), HEPES (1x, Gibco), PenStrep (1x, Gibco), 50% WNT3A-, 20% RSPO1-, and 10% NOG(gin)-conditioned medium (all homemade), 2% B27 with vitamin A (Gibco), 1.25 mM *N*-Acetylcysteine, 10 mM Nicotinamide, 50 ng/ml murine-EGF (Peprotech), 500 nM A83-01, and 10 mM

SB202190 (Sigma). The medium was changed every 2–4 days and organoids were passaged 1:4–1:8 each week. After thawing, organoids were passaged at least once before electroporation.

**Cell culture, lentiviral production, and lentiviral transduction**. HEK293T, Caco-2 and HeLa cells were obtained from the ATCC.

Fibroblasts were maintained and split every 7 days in standard medium, consisting of F-12 Nut Mix (Ham) (Gibco), 10% FBS (Gibco), and PenStrep (1x, Gibco). HEK293T, Caco-2, and HeLa cells were maintained and split every 4-5 days in standard medium, consisting of DMEM + GlutaMAX (1x, Gibco), 10% FBS (Gibco), and PenStrep (1x, Gibco). For production of lentivirus, HEK293T cells were plated in a 145 mm CELLSTAR dish (Corning) in standard medium without PenStrep and transfected 24 hours later (at 50–60% confluence) with a mix of 10 μg of the pLenti-CMV-GFP-Hygro plasmid, 5 μg of psPAX2, 5 μg of pMD2.G, and 60 μl of polyethylenimine (1 mg/ml). pLenti-CMV-GFP-Hygro was a gift from Eric Campeau & Paul Kaufman (Addgene #17446). psPAX2 and pMD2.G were gifts from Didier Trono (Addgene plasmids #12260 and #12259). 24 h after transfection, the medium was replaced with standard medium with PenStrep, and virus-containing medium was harvested at 48 and 96 hours after transfection. Medium was centrifuged at 400 × g for 5 min and supernatants were filtered through a 0.22-μm filter, after which virus particles were concentrated by ultracentrifugation at 50,000 × g for 2 hours and resuspension in 1 ml DMEM. HEK293T cells were transduced at an MOI of 2 for 24 h in the presence of polybrene (8 μg/μl) and analyzed by FACS 14 days after transduction to confirm stable GFP expression.

**Transfection of HEK293T cells**. HEK293T cells were plated in standard medium at a density of 50,000 cells per well in a 96-well plate 1 day prior to transfection. HEK293T cells were transfected with 0.1 μg fluoPEER plasmid, 0.25 μg prime editor plasmid, 0.1 μg pegRNA plasmid, and optionally 0.05 μg nicking-gRNA plasmid in a mix of 25 μl OptiMEM and 0.5 μl lipofectamine 2000 for each well.

**Electroporation of organoid cells**. Before electroporation, organoids were grown under standard culture conditions. Four wells containing organoids were then dissociated for each condition using TrypLE (Gibco) for 4–5 min at 37 °C, after which mechanical disruption was applied through pipetting. Cells were washed once using Advanced DMEM/F12, resuspended in 80 μl OptiMEM containing Y-27632 (10 μM), and 20 μl DNA mixture was added. For prime editing, the DNA mixture contained 4 μg fluoPEER, 12 μg prime editor plasmid, 4 μg pegRNA plasmid, and 2 μg nicking sgRNA plasmid. The cell-DNA mixture was transferred to an electroporation cuvette and electroporated using a NEPA21 electroporator (NEPA GENE) with 2× poring pulse (voltage: 175 V, length: 5 ms, interval: 50 ms, polarity: +) and 5× transfer pulse (voltage: 20 V, length: 50 ms, interval: 50 ms, polarity ± ), as previously described[37]. Cells were removed from the cuvette and transferred into 500 μl OptiMEM containing Y-27632 (10 μM). After 20 minutes, cells were plated in 120 μl matrigel divided over four wells. Upon polymerization of the Matrigel, hl-EM or hSI-EM was added containing Y-27632 (10 μM).

**Transfection of fibroblast and HeLa cells**. Skin-derived fibroblasts were grown under standard culture conditions and plated on 12-well plates 3 days prior to transfection so that confluency was 60–70% at transfection. HeLa cells were grown under standard culture conditions and plated on 24-well plates 1 day prior to transfection so that confluency was 60–70% at transfection. For prime editing experiments, fibroblasts and HeLa cells were transfected with 0.12 μg fluoPEER plasmid, 0.19 μg prime editor plasmid, 0.05 μg pegRNA plasmid and 0.05 μg nicking-gRNA plasmid in a mix of 8.4 μl OptiMEM, 0.6 μl lipofectamine 3000, and 0.4 μl P3000 reagent. For base editing experiments, the DNA mix consisted of 0.14 μg fluoPEER plasmid, 0.2 μg ABE8e-TadA (V106W) plasmid, and 0.06 μg sgRNA. ABE8e (TadA-8e V106W) was a gift from David Liu (Addgene #138495).

**FACS**. Organoids and cell lines were harvested and dissociated to single cells using TrypLE (Gibco) or Trypsin (Gibco), respectively, after which cells were resuspended in FACS buffer (phosphate-buffered saline with 2 mM ethylenediamine-tetraacetic acid and 0.5% bovine serum albumin). Prior to FACS, cells were filtered through a 5 ml Falcon polystyrene test tube (Corning). Flow cytometry was performed on the FACS Fortessa (BD) and sorting was performed on the FACS FUSION (BD) using FACS Diva software (BD). Sorted cells were collected in culture medium and spun down. Gating strategy for cells included for fluoPEER analysis is shown in Supplementary Fig. 12a. The ratio was calculated by dividing the average measured Cherry signal by the average measured GFP signal for all GFP+ cells.

**Genotyping**. Sorted cells were harvested using the Quick-DNA microprep kit (Zymogen) according to manufacturer's protocols. PCR was performed on the genomic region of interest using the Phusion polymerase (ThermoFisher) or Q5 polymerase (NEB) and purified using the QIAquick PCR Purification Kit (Qiagen) according to manufacturer instructions. The PCR product was sent for Sanger

sequencing to EZSeq Macrogen Europe. All PCR and sequencing primers used are listed in Supplementary Data file 2.

**High-throughput DNA sequencing of genomic DNA samples.** Genomic sites of interest were amplified from genomic DNA samples and sequenced on an Illumina iSeq 100 as previously described[38]. In short, PCR primers containing Illumina forward and reverse adapters (Supplementary Data file 2) were used in a first amplification reaction (PCR1) of 25 µl using Q5 polymerase (NEB) to amplify the genomic region of interest. In a second round of PCR (PCR2, 25 µl), 1 µl of each PCR1 is barcoded with unique Truseq DNA Index primers (Illumina) and isolated from gel. DNA concentration was measured by fluorometric quantification (Qubit, ThermoFisher Scientific) and sequenced on an Illumina iSeq 100 instrument according to the manufacturer's protocols to create 2 × 150 bp paired-end reads. The resulting FASTQ files were analyzed with the RGEN PE-analyzer, using the unedited sequence as the reference sequence and the prime-edited sequence as the intended sequence[39]. Prime editing efficiency was calculated as the percentage of (RGEN PE-reads/RGEN more than minimum frequency reads). For unwanted byproduct analysis at the pegRNA or nickase sgRNA site, a comparison range (R) of 30 bp or 70 bp was used so that 60 bp or 140 bp flanking the predicted nicking site were considered. Frequency of indels was calculated as the percentage of (RGEN reads with unwanted inserts and deletions/RGEN more than minimum frequency reads).

**RNA sequencing.** HEK293T cells were transfected with fluoPEER, PE2, pegRNA, and nicking sgRNA plasmids and FACS sorted after 48 h. Total RNA was isolated using Trizol LS reagent (Invitrogen) and stored at –80 °C until further processing. mRNA was isolated using Poly(A) Beads (NEXTflex). Sequencing libraries were prepared using the Rapid Directional RNA-Seq Kit (NEXTflex) and sequenced on a NextSeq500 (Illumina) to produce 75 base long reads (Utrecht DNA Sequencing Facility). Sequencing reads were mapped against the reference genome (hg19 assembly, NCBI37) using BWA[40] package (mem –t 7 –c 100 –M –R). Raw reads were further analyzed as described under 'Data analysis'.

**Chemical cell cycle synchronization.** Chemical cell cycle synchronization using thymidine and nocodazole was performed as described previously[41,42]. In short, 50,000 HEK293T cells or 10,000 Caco-2 cells were plated in 24-well plates. After 8 h, 2 mM thymidine was added to the cells. 17 h later, cells were washed twice and medium was replaced with standard culture medium. 8 h later, 2 mM thymidine was readded to the cells. At the same time point, 200 ng/ml nocodazole was added for the nocodazole treatment. 20 h later, cells were washed twice, medium was replaced with standard culture medium, and prime editing was performed by transfection of 0.25 µg prime editor plasmid and 0.1 µg pegRNA plasmid in a mix of 100 µl OptiMEM and 0.3 µl lipofectamine 2000 for each well. 48 h later, transfected cells were sorted using flow cytometry and genotyping was performed as described above.

**fluoPEER cell cycle analysis.** 200 ng/ml nocodazole was added to HEK293T cells 20 h before FACS analysis as a control for cells in G2 phase. 24 h before fluoPEER read-out and genomic Sanger sequencing, HEK293T cells were transfected. 60 min before harvesting for FACS analysis, 10 µg/ml Hoechst 33342 (ThermoFisher) was added to the culture medium. Gating strategy for G1, S, and G2 phases of the cell cycle is shown in Supplementary Fig. 12b.

**Data analysis.** Flow cytometry data were analyzed using FlowJo™ Software. RNA sequencing was analyzed using DESeq2 in RStudio[43], gene set enrichment analysis[44], and enrichR[45]. All figures were made in Prism (GraphPad Software) or GGPlot2[46] in RStudio. Sanger sequencing was quantified using EditR[47] or Tide[48]. Sanger sequencing chromatograms were made in Benchling. NGS data were quantified and analyzed using RGEN PE-analyzer[39].

**Statistics and reproducibility.** No pre-specified effect size was calculated, and no statistical method was used to predetermine sample size. For comparisons of multiple groups, an ordinary one-way ANOVA with Holm–Sidak correction for multiple comparisons was used and performed in Prism (GraphPad Software). Statistical tests were appropriate for comparisons being made; assessment of variation was carried out but not included. Experiments were not randomized. Reproducibility: Fig. 1c representative of three biologically independent replicates from one experiment. In Fig. 1d, each point in the dot-plot represents the mean of three (reporter rank) or two (genomic rank) replicates in two independent experiments. Each point in graphs of Fig. 1e represents the mean of at least two biologically independent replicates for each prime editing condition in a single experiment. Figure 1f representative of (1) 14/20 clonally picked intestinal organoids from two different patients with biallelic CFTR^F508del mutations, that showed swelling after addition of 1 µM forskolin to the medium, (2) 2/10 clonally picked liver-derived organoids from a patient with biallelic ABCB4^E1012X mutations, (3) 3/10 clonally picked intestinal organoids from a patient with biallelic ATP8B1^R600Q mutations, and (4) 2/10 clonally picked liver-derived organoids from a healthy

control, in which biallelic IARS1^I1174N were created; 4/10 clonally picked organoids from the same experiment showed monoallelic IARS1^I1174N mutations.

Figure 2a is representative of two replicates in two independent experiments. In Fig. 2b each dot represents the ratio of two bars (PE2* vs. PE2) in Supplementary Fig. 8a, b. In Supplementary Figure 8a, b, each bar represents the mean of 2–3 biologically independent replicates. Figure 2c, d, e are representative of three biologically independent replicates from one experiment. Figure 2f shows representative data from one experiment. Figure 3a, b are representative of at least two biologically independent replicates from one experiment. Data from Fig. 3c is based on one transfection, of which 8 conditions of 100 GFP + cells and 2 conditions of 100 GFP + RFP + cells were sorted. Figure 3d is representative of three biologically independent replicates from one experiment. Figure 3e is representative of two or three biologically independent replicates from one experiment. Figure 4b–d are data from RNA sequencing of two biologically independent replicates for both conditions (GFP + RFP-/GFP + RFP +) for each edit (CTNNB1/IARS1) from one experiment. In Fig. 4e HEK293T, each dot represents the mean of three biologically independent replicates, three independent experiments were performed. In Fig. 4e Caco-2, each dot represents a biologically independent replicate from one experiment.

Supplementary Figure 4 is representative of two replicates from two independent experiments. The fluoPEER data in Supplementary Fig. 5a, b are representative of three biologically independent replicates in one experiment; fluoPEER data in supplementary Fig. 5c is representative of two biologically independent replicates in one experiment that was characteristic of two repeated experiments. Supplementary Fig. 6a–c are representative of two biologically independent replicates per condition in one experiment; for representation of the percentage of GFP+ cells, data from different conditions were pooled. Supplementary Fig. 6d is based on data from three biologically independent replicates of sorted cells to seed 100% GFP + conditions. Each pegRNA-PE2-fluoPEER combination from Supplementary Fig. 7 was tested in at least two independent transfection experiments. Supplementary Figure 7 is representative of two or three biologically independent replicates from one characteristic experiment. Supplementary Figure 7b, c representative of clonally picked liver organoids with monoallelic (two clones) or biallelic (two clones) IARS1^I1174N mutations. Supplementary Fig. 7d representative of 20 clonally picked liver-derived organoids from a patient with biallelic MUT^R329H mutations. Supplementary Figure 8c is representative of three biologically independent replicates from one experiment. Supplementary Figure 9 shows FACS plots of single conditions that are representative of two independent transfection experiments. Supplementary Figure 10a–d represent data from two biologically independent replicates for each experimental group. Supplementary Figure 10e represents data from three biologically independent replicates for two independent biological replicates. Supplementary Figure 11 data is representative of three biologically independent replicates from NGS. Supplementary Figure 12 is representative FACS data for all FACS experiments shown.

**Reporting summary.** Further information on research design is available in the Nature Research Reporting Summary linked to this article.

## Data availability
Source data are provided with this paper. The RNA sequencing data generated in this study have been deposited in the National Center of Biotechnology Information (NCBI) database under accession code GSE195977. High-throughput sequencing data have been deposited at the NCBI Sequence Read Archive database at PRJNA802707. All plasmids created for this study are available upon reasonable request. Source data are provided with this paper.

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

## Acknowledgements

The authors thank the Kim lab of the Department of Pharmacology, Yonsei University College of Medicine for supplying the PE_prediction code. This work was supported by Metakids funding (to S.A.F.), a Clinical Fellows grant from The Netherlands Organisation for Health Research and Development Health Institute (40-00703-97-13537 to S.A.F.), funding from Stichting Reggeborgh (to S.A.F. and R.H.J.H.), and a Boost Grant Child Health from the WKZ, UMC Utrecht (to I.F.S.).

## Author contributions

I.F.S., I.P.J., and S.A.F. designed the project; J.H.L.B. analyzed the pathogenic mutations in the ClinVar database using PE_prediction; H.P.J.D. and R.H.J.H. helped establish the biobank of patient-derived stem cell organoids used in this study; I.F.S., I.P.J., R.S., G.K., S.S., E.F.I., D.P.B., and R.C.M.R performed experiments and analyses; S.A.S. and J.M.B. provided the cystic fibrosis samples used in this study; I.F.S., I.P.J., E.E.S.N., and S.A.F. wrote the manuscript.

## Competing interests

The authors declare no competing interests.
