## [Peer Review File · Nature Communications]

Reviewers' Comments:

Reviewer #1:

Remarks to the Author:

In this manuscript, Schene and colleagues describe the development of a reporter system entitled fluorescent prime editing and enrichment reporter (fluoPEER) that can be employed to evaluate the efficiency of pegRNA design and prime editor efficiency as well as enrich for prime edited cell populations. In addition, the authors apply this tool to investigate the mechanisms that are involved in regulating prime editing efficiency. Overall, although the authors present a potentially useful tool and some mechanistic insight into prime editing, the manuscript in its current form suffers from some major shortcomings that need to be addressed before publication.

1. Introduction and Discussion

a. The authors only highlight a few of the previous base-reporter editing systems but there are several others that were developed before or along with the ones referenced (e.g. BMC Biol . 2018 Dec 28;16(1):150; Sci Rep. 2019 Jan 24;9(1):497; Mol Ther. 2020 Jul 8;28(7):1696-1705.; Nucleic Acids Res. 2019 Nov 4;47(19):e120). A discussion of these should be included.

b. What are the advantages and disadvantages of FluoPeer compared to other prime editing enrichment strategies that have been conventionally employed (e.g. reporters of transfection or expression) or pre-published (e.g. PEAR, <https://www.biorxiv.org/content/10.1101/2021.04.26.441486v1.full>).

2. Figure 1 and related Supplementary Figures.

a. The authors describe how fluoPEER can be used for efficient pegRNA design. However, in the current form this still requires that each potential target site be cloned into the fluoPEER reporter and only eliminates a single step in conventional analysis of pegRNA design (i.e. Sanger sequencing or NGS of edited cell populations). The authors should comment on this in their discussion. Along similar lines, to demonstrate the utility of this tool in pegRNA design the authors should provide a more thorough comparison of fluoPEER versus bioinformatic predication algorithms than which is currently presented in Supplementary Figure 5. More specifically, the authors should provide comparisons at more than 3 target sites and in the context of different types of edits (i.e. different types of single base substitutions, insertions) as well as against different prediction algorithms in addition to DeepPE (e.g. PrimeDesign, Nat Commun .2021 Feb 15;12(1):1034.; multicrispr, Life Sci Alliance. 2020 Sep 9;3(11):e202000757; PINE-CONE, ACS Synth Biol. 2021 Feb 19; 10(2): 422-427.; pegFinder, Nat Biomed Eng. 2021 Feb;5(2):190-194.)

b. In Figure 1e, the authors evaluate the efficiency of various pegRNA designs at multiple loci. However, the efficiency is only evaluated in the context of the episomal reporter plasmid and not at the corresponding genomic site. In other words, what is the correlation between relative Cherry/GFP and the actual editing at the genomic target site as evaluated by sequencing of edited cell populations?

3. Figure 2 and related Supplementary Figures. The authors use the FluorPeer system to characterize new PE2 variants (Figure 2a-c) as well as the PE3b (Figure 2d) system. Again, this evaluation is only performed on the episomal reporter plasmid (Figure 2a-c) and a limited number of target sites (Figure 2d). To what extent do these findings correlate to editing at genomic loci?

4. Figure 3 and related Supplementary Figure. In Figure 3, the authors demonstrate how FluoPEER can be used to enrich for genomic editing at a limited number of sites. However, this analysis in its current form is very limited as described below.

a. It is becoming more common for labs to use reporters of expression to allow for enrichment of prime-editing cells (e.g. PE2-SpG-P2A-GFP; Kweon et al Mol Ther. 2021). How does enrichment using FluoPEER compare to these methods?

b. Does enrichment of editing in the GFP+Cherry+ cells also correspond to an increase in off-target editing as well and indel formation?

c. The conventional standard for analyzing editing outcomes is NGS analysis of the targeted locus to evaluate individual allelic outcomes. The use of bulk sequencing and editing efficiency as the authors perform does not provide this level of detail.

d. In many applications, for FluoPEER to be a useful tool will require the downstream establishment of clonal cell populations at higher efficiencies than current approaches. Does

FluoPEER enable this in terms of cell survival and transience of the reporter system?

5. Figure 4 and related Supplementary Figures.

a. The authors perform RNA-seq analysis to identify genes that are differentially expressed in the editing cell populations. Please include the entire list of differentially expressed genes in a Supplementary Table

b. Knockouts of 7 genes that were enriched in the RNA-seq analysis were performed. How were these 7 genes prioritized? Would overexpression of genes increase prime editing efficiency?

c. The outcomes of only 2 of the 7 genes knocked down were provided. What were the outcomes from the other gene knockdown experiments?

Reviewer #2:

Remarks to the Author:

This paper describes the construction and use of a tool for optimizing the efficiency of prime editing for targeting sites of interest in DNA. Based on my reading of the existing prime editing literature, prime editing has extremely variable efficiency making its use difficult to the extent that editing each individual site of interest is almost a research project. I should further add that the description of prime editing vectors is not exactly useful either. In this study, the authors develop a tool consisting on an upstream GFP reporter and a downstream Cherry reporter between which is cloned a target region containing a stop codon (either natural or introduced); the construct is called fluoPEER. Prime editing of a site within the target is designed to co-correct the stop codon. Thus, correct targeting increases the Cherry to GFP expression ratio, which can then be monitored in order to guide the optimization of the targeting constructs. The use of this tool is then illustrated through a large number of editing experiments. My overall view is that this approach is novel, well-demonstrated and useful, and that it would be a useful addition to the literature that would help people who are interested in using prime editing.

I have a number of concerns about this paper, and in particular how difficult it is for non-experts to read. It is written in too abbreviated a way. It is loaded with CRISPR slang that not everyone may be familiar with. It contains numerous minor errors and omissions. And it highlights some experiments that seem under-developed and not that meaningful at their present stage of development. The utility of the paper could be improved with some re-writing and editing. I make a few specific comments below but cannot cover everything.

Examples of errors and omissions. In the second paragraph of the results the authors state that editing could only be detected in conditions with >10% BFP+ cells but in the relevant figure legend they state that the limit is >5%. The Y-axis label of Supp Fig 4C could use some explanation (CRISPR slang that will lose the general reader). In the third paragraph of the results the authors claim that fluoPEER ranked the efficiencies correctly but in Supp Fig 5 Target 3 pegRNA2 the prediction is not correct and its not that clear which method one would pick for Target 2 predictions. Another example is the Methods where in virtually every method involving growth of cells, the authors fail to list the medium used to initially grow the cells only to have the name of some medium used in the procedure appear in the middle of the description creating confusion about what medium is used when. And what are CTN and IARS in Supp Fig 11C abbreviations for? I offer these as examples that could be looked at, but note they are not the only examples.

I found the experiments on cell cycle regulation and gene expression/DNA repair to be superficial and think they should be removed from the paper. With regard to cell cycle regulation, the authors only document a small effect of different phases of the cell cycle on targeting. Indeed, would anyone be motivated by the data in Fig 4F to collect cells at G1/S to use to improve the efficiency of editing? I suspect not. In the realm of only publishing the most useful experiments, this is an experiment that could be eliminated. Similarly, I found the experiments relating to DNA repair to be somewhat unsatisfying. It isn't clear what the exact magnitude of either increased or decreased expression is or whether this is functionally significant. It is also not clear how the gene list in Supp Fig 11C was derived and whether it represents all of the repair genes implicated or if it's a partial set. If it's the entire list, it seems like a random list of repair genes rather than what one would observe if β n specific repair pathways were important. I don't think knocking out selected

genes addresses functional significance of gene expression patterns. Analysis of 7 genes in a diversity of repair pathways doesn't seem either comprehensive or that informative. The finding of PMS1 is unusual as it plays such a minor role in DNA repair. Furthermore, since true quantitative mutation rates are not being determined, its not clear how prone to accumulating deletions the BRCA2 and PMS1 mutants are. Basically, the authors are presenting a one-off observation on a few DNA repair genes as compared to a more useful, comprehensive analysis of the effect of DNA repair defects on prime editing. Like the cell cycle analysis, the DNA repair analysis adds little at this stage of the study.

Response to reviewer #1

In this manuscript, Schene and colleagues describe the development of a reporter system entitled fluorescent prime editing and enrichment reporter (fluoPEER) that can be employed to evaluate the efficiency of pegRNA design and prime editor efficiency as well as enrich for prime edited cell populations. In addition, the authors apply this tool to investigate the mechanisms that are involved in regulating prime editing efficiency. Overall, although the authors present a potentially useful tool and some mechanistic insight into prime editing, the manuscript in its current form suffers from some major shortcomings that need to be addressed before publication.

We thank reviewer #1 for the thorough and critical reading of our manuscript, which helped us to strengthen our data and increase the applicability of fluoPEER for other groups. In general, we added supporting data by running Next Generation Sequencing (NGS) experiments for reporter validation and enrichment as suggested by the reviewer. We find that fluoPEER performs very well in predicting prime editing efficiency when compared to NGS quantification, at various additional sites, performing different types of mutations. Furthermore, our comparison to DeepPE has now been strengthened by adding more sites in different contexts. Considering recent developments in the field of prime editing, we added engineered pegRNAs (epgRNAs) and a novel prime editing variant (PE4max) to our validations, finding that the effect of these developments on prime editing efficiency can be tested using fluoPEER. As the reviewer rightly pointed out, important information was missing from our discussion and introduction. We added the necessary literature and mentioned important points as per request.

1. Introduction and Discussion

a. The authors only highlight a few of the previous base-reporter editing systems but there are several others that were developed before or along with the ones referenced (e.g. BMC Biol . 2018 Dec 28;16(1):150; Sci Rep. 2019 Jan 24;9(1):497; Mol Ther. 2020 Jul 8;28(7):1696-1705.; Nucleic Acids Res. 2019 Nov 4;47(19):e120). A discussion of these should be included.

Indeed the base-reporter editing systems are important to mention in this manuscript and all relevant systems have been added to the introduction: “Previously developed genome editing reporters for ... Wang et al., 2020).”

b. What are the advantages and disadvantages of FluoPeer compared to other prime editing enrichment strategies that have been conventionally employed (e.g. reporters of transfection or expression) or pre-published (e.g. PEAR, <https://www.biorxiv.org/content/10.1101/2021.04.26.441486v1.full>).

The reviewer is correct in stating that this was missing in our manuscript and this has been added to the discussion: “The Prime Edit Activity Reporter (PEAR) depends ... a genome-targeting pegRNA (Simon et al., 2021, figure 3d).”

2. Figure 1 and related Supplementary Figures.

a. The authors describe how fluoPEER can be used for efficient pegRNA design. However, in the current form this still requires that each potential target site be cloned into the fluoPEER reporter and only eliminates a single step in conventional analysis of pegRNA design (i.e. Sanger sequencing or NGS of edited cell populations). The authors should comment on this in their discussion. Along similar lines, to demonstrate the utility of this tool in pegRNA design the authors should provide a more thorough comparison of fluoPEER versus bioinformatic predication algorithms than which is currently presented in Supplementary Figure 5. More specifically, the authors should provide comparisons at more than 3 target sites and in the context of different types of edits (i.e. different types of single base substitutions, insertions) as well as against different prediction algorithms in addition to DeepPE (e.g. PrimeDesign, Nat Commun .2021 Feb 15;12(1):1034.; multicispr, Life Sci Alliance. 2020 Sep 9;3(11):e202000757; PINE-CONE, ACS Synth Biol. 2021 Feb 19; 10(2): 422–427.; pegFinder, Nat Biomed Eng. 2021 Feb;5(2):190-194.)

We agree with the reviewer that our comparison to DeepPE was not thorough enough, so we have added more sites, which strengthens our findings. We could not include comparisons in the contexts of different types of edits, since the DeepPE algorithm only provides prediction scores for pegRNAs with different PBS and RTT lengths for a G>C substitution at position +5. Still, we do validate the use of fluoPEER for the prediction of substitutions and insertions in Fig. 1e. With regards to PrimeDesign, PINE-CONE, and pegFinder, we did not include those algorithms in our comparisons, as these algorithms are not based on data-driven prediction. Rather, these algorithms rely on set values that output pegRNAs based on current knowledge of pegRNAs. As such, we consider these algorithms to be pegRNA design tools rather than genuine prediction algorithms, and therefore inferior to DeepPE. We do agree however that these design tools should be mentioned and we discuss them in the updated version of the introduction: “A number of pegRNA design tools have been developed ... (Hsu et al., 2021; Standage-Beier et al., 2021; Chow et al., 2021).”

b. In Figure 1e, the authors evaluate the efficiency of various pegRNA designs at multiple loci. However, the efficiency is only evaluated in the context of the episomal reporter plasmid and not at the corresponding genomic site. In other words, what is the correlation between relative Cherry/GFP and the actual editing at the genomic target site as evaluated by sequencing of edited cell populations?

We have added new data to validate the correlation between editing on the episomal fluoPEER reporter and the corresponding genomic site for three different additional genomic sites and types of mutations in Fig. 1e. The data originally shown in Fig. 1e has been moved to Supplementary Fig. 7. However, the prime edits performed in this new Supplementary Fig. 7 all correct a pathogenic mutation in the context of a patient genome. Since we only have organoid cells for many of these mutations, and given the difficulty of performing many different transfection conditions in organoid cells, we only performed the optimally predicted prime edit strategy in patient organoid cells, yielding the results shown in Fig. 1f.

3. Figure 2 and related Supplementary Figures. The authors use the FluorPeer system to characterize new PE2 variants (Figure 2a-c) as well as the PE3b (Figure 2d) system. Again, this evaluation is only performed on the episomal reporter plasmid (Figure 2a-c) and a limited number of target sites (Figure 2d). To what extent do these findings correlate to editing at genomic loci?

We have added data to elucidate the points mentioned above (Fig. 1e, Fig. 2c), as well as added epegRNA data (Fig. 2d) and PE4max data (Supplementary Fig. 8c). Note that for the data in Fig. 2b, the right panel represents relative editing efficiency on the genomic DNA. Note that for Fig. 1e, 2c, 2d, and 2e, fluoPEER prediction scores are all compared to genomic editing, as requested by the reviewer, as quantified by next-generation sequencing (NGS).

4. Figure 3 and related Supplementary Figure. In Figure 3, the authors demonstrate how FluoPEER can be used to enrich for genomic editing at a limited number of sites. However, this analysis in its current form is very limited as described below.

a. It is becoming more common for labs to use reporters of expression to allow for enrichment of prime-editing cells (e.g. PE2-SpG-P2A-GFP; Kweon et al Mol Ther. 2021). How does enrichment using FluoPEER compare to these methods?

Since we agree with the reviewer, we have decided to compare total transfected cells (i.e. fluoPEER-GFP+ cells) to fluoPEER-edited cells (i.e. GFP+RFP+ cells). However, we did not compare with another expression-reporter plasmid, since none such data has been published yet. Furthermore, the co-transfection data presented in Supplementary Fig. 9 indicate that the fluoPEER-GFP+ population is essentially the same as the ‘PE2-plasmid+’ population. Finally, we feel that the fairest comparison within one experiment is to draw from the same cell population and compare editing efficiencies within those conditions. As such, experimental noise is kept at a minimum and true differences can be quantified.

b. Does enrichment of editing in the GFP+Cherry+ cells also correspond to an increase in off-target editing as well and indel formation?

We are thankful for this suggestion and added the indel formation data to our manuscript (Fig. 2b). In line with previous reports of very low off-target effects of prime editing, we could not identify any off-target effects to report in our edits.

c. The conventional standard for analyzing editing outcomes is NGS analysis of the targeted locus to evaluate individual allelic outcomes. The use of bulk sequencing and editing efficiency as the authors perform does not provide this level of detail.

We have transformed all important data to NGS for the support of fluoPEER findings (Fig. 1e, 2c, 2d, 2e, 3a, and 3b). As mentioned above, this allowed analysis of low-frequency allelic outcomes, including unwanted indel formation, as shown in Fig. 3b.

d. In many applications, for FluoPEER to be a useful tool will require the downstream establishment of clonal cell populations at higher efficiencies than current approaches. Does FluoPEER enable this in terms of cell survival and transience of the reporter system?

We thank the reviewer for this suggestion and added organoid data to show that fluoPEER supports the growth of clonal organoid lines (Fig. 3c). Furthermore, we added data showing that fluoPEER GFP-positivity diminishes quickly after 10 days (Supplementary Fig. 6d).

5. Figure 4 and related Supplementary Figures.

a. The authors perform RNA-seq analysis to identify genes that are differentially expressed in the editing cell populations. Please include the entire list of differentially expressed genes in a Supplementary Table

We have provided all differentially expressed genes in Supplementary Table 2, as suggested.

b. Knockouts of 7 genes that were enriched in the RNA-seq analysis were performed. How were these 7 genes prioritized? Would overexpression of genes increase prime editing efficiency?

See the answer to c. below.

c. The outcomes of only 2 of the 7 genes knocked down were provided. What were the outcomes from the other gene knockdown experiments?

Based on the comments of reviewer #2 and given the recently published study of Chen et al., 2021 (Enhanced prime editing systems by manipulating cellular determinants of editing outcomes), we have removed this data from the manuscript. Still, we would like to note that Chen et al. (2021) do find similar importance of homologous recombination-related genes in preventing indel formation by prime editing. We mention this in the discussion.

Response to reviewer #2

This paper describes the construction and use of a tool for optimizing the efficiency of prime editing for targeting sites of interest in DNA. Based on my reading of the existing prime editing literature, prime editing has extremely variable efficiency making its use difficult to the extent that editing each individual site of interest is almost a research project. I should further add that the description of prime editing vectors is not exactly useful either. In this study, the authors develop a tool consisting on an upstream GFP reporter and a downstream Cherry reporter between which is cloned a target region containing a stop codon (either natural or introduced); the construct is called fluoPEER. Prime editing of a site within the target is designed to co-correct the stop codon. Thus, correct targeting increases the Cherry to GFP expression ratio, which can then be monitored in order to guide the optimization of the targeting constructs. The use of this tool is then illustrated through a large number of editing experiments. My overall view is that this approach is novel, well-demonstrated and useful, and that it would be a useful addition to the literature that would help people who are interested in using prime editing.

We are thankful for the feedback of reviewer #2, which helped us to improve the readability of our manuscript and focus more on our important data. We are happy to hear that our approach is considered to be a useful addition to literature and helpful for people who are interested in using prime editing, as that is why we set out to develop fluoPEER in the first place. Due to the comment regarding publishing the most useful experiments, we removed the DNA repair gene knock-out experiments. We agree that this data is too divergent and not developed sufficiently for publication. Our cell cycle data we deemed important enough for various reasons, so we supported our findings by adding new data in Caco-2 cells. Recent publications have supported the hypothesis that the cell cycle affects prime editing efficiency. As such, we think it's important to add our cell cycle data to further develop this line of thinking in literature. By adding existing literature about prime editing and the cell cycle to our discussion, we have tried to put our data in the right scientific context. Lastly, as per request, we adjusted our manuscript in various places to remove inconsistencies and increase the comprehensibility of CRISPR-related expressions.

I have a number of concerns about this paper, and in particular how difficult it is for non-experts to read. It is written in too abbreviated a way. It is loaded with CRISPR slang that not everyone may be familiar with. It contains numerous minor errors and omissions. And it highlights some experiments that seem under-developed and not that meaningful at their present stage of development. The utility of the paper could be improved with some re-writing and editing. I make a few specific comments below but cannot cover everything.

Examples of errors and omissions. In the second paragraph of the results the authors state that editing could only be detected in conditions with >10% BFP+ cells but in the relevant figure legend they state that the limit is >5%. The Y-axis label of Supp Fig 4C could use some explanation (CRISPR slang that will lose the general reader). In the third paragraph of the results the authors claim that fluoPEER ranked the efficiencies correctly but in Supp Fig 5 Target 3 pegRNA2 the prediction is not correct and its not that clear which method one would pick for Target 2 predictions. Another example is the Methods where in virtually every method involving growth of cells, the authors fail to list the medium used to initially grow the cells only to have the name of some medium used in the procedure appear in the middle of the description creating confusion about what medium is used when. And what are CTN and IARS in Supp Fig 11C abbreviations for? I offer these as examples that could be looked at, but note they are not the only examples.

We thank the reviewer for carefully reading our manuscript and highlighting some of the inconsistencies which we have overlooked. We have adjusted these and other errors in order to improve the consistency and readability of our manuscript. Furthermore, we took care to remove CRISPR-related slang and explained more of the techniques and expressions in the introduction.

I found the experiments on cell cycle regulation and gene expression/DNA repair to be superficial and think they should be removed from the paper. With regard to cell cycle regulation, the authors only document a small effect of different phases of the cell cycle on targeting. Indeed, would anyone be motivated by the data in Fig 4F to collect cells at G1/S to use to improve the efficiency of editing? I suspect not. In the realm of only publishing the most useful experiments, this is an experiment that could be eliminated. Similarly, I found the experiments relating to DNA repair to be somewhat unsatisfying. It isn't clear what the exact magnitude of either increased or decreased expression is or whether this is functionally significant. It is also not clear how the gene list in Supp Fig 11C was derived and whether it represents all of the repair genes implicated or if it's a partial set. If it's the entire list, it seems like a random list of repair genes rather than what one would observe if $\beta\pi$ specific repair pathways were important. I don't think knocking out selected genes addresses functional significance of gene expression patterns. Analysis of 7 genes in a diversity of repair pathways doesn't seem either comprehensive or that informative. The finding of PMS1 is unusual as it plays such a minor role in DNA repair. Furthermore, since true quantitative mutation rates are not being determined, it's not clear how prone to accumulating deletions the BRCA2 and PMS1 mutants are. Basically, the authors are presenting a one-off observation on a few DNA repair genes as compared to a more useful, comprehensive analysis of the effect of DNA repair defects on prime editing. Like the cell cycle analysis, the DNA repair analysis adds little at this stage of the study.

These critical comments of the reviewer have helped us re-evaluate our manuscript and data. We agree that the DNA repair gene knock-out data was insufficiently supported and we removed the figures from the manuscript. For the cell cycle data, we agree that chances are small that researchers will use a thymidine block on cells to increase prime editing efficiency. However, we hope that adding this data to our manuscript will increase understanding of the impact of cellular mechanisms on prime editing. Recent reports have underlined the role of the cell cycle and DNA repair genes in prime editing. More specifically, one report shows that prime editing is 1.5x higher in cycling vs. non-cycling cells (Wang et al., 2021). Moreover, a large CRISPR interference screen has been published in order to uncover important cellular mechanisms underlying prime editing (Chen et al., 2021). We find many of the same genes important in prime editing, supporting our hypotheses. As such, we decided to add data of prime editing in Caco-2 cells stalled at the G1/S and G2/M boundaries, showing that G1/S stalling again increases prime editing efficiency. To support our data, we have provided additional literature in the discussion "The cellular mechanisms ... increased successful editing." and comparisons to our data (Fig. 4d).

Reviewer #1 (Remarks to the Author):

In this revised manuscript, Schene and colleagues address many (but not all) of the critiques of the previous reviews. However, there are some unaddressed concerns that still remain that should be addressed. Also, given that the authors have removed what was considered potentially the most impactful aspect of the manuscript, it is not clear if this manuscript in its current form sufficiently advances the field to warrant publication in Nature Communications.

(1) I still disagree with the authors' assertion that their system should not be compared to reporters of expression (e.g. PE2-SpG-P2A-GFP; Kweon et al Mol Ther. 2021). While it is appreciated that the authors have now made the comparisons between all transfected cells versus fluor PEER-enriched cells, the comparison to cells in which the prime editors are expressed is still a critical one to validate the impact of their reporter system. In other words, enriching cells in which the prime editor is expressed may result in similar levels of editing to that of cells in which the fluor PEER reporter is active. If this were the case, then successful prime editing would not require the cloning and optimization for each site that is required for the fluor PEER system but instead simply enriching for cells in which the prime editor is being actively expressed. Along similar lines, the cellular mechanisms that authors described affecting prime editing might affect expression of the prime editor and not the prime editor's ability to modify genomic loci. However, without this important comparison this cannot be determined.

(2) As stated in the previous review, the authors should include information about indel information and off-target effects in the fluor-peer enriched cells versus the all transfected cells. If enriching for editing is also enriching for indels and off-targets, then this is a severe limitation of the system that the authors do not discuss.

(3) It was disappointing to see that the authors removed the shRNA data rather than addressing the previous critiques of the reviewers. This was considered the most impactful component of the manuscript as methods to improve prime editing efficiency in the absence of reporter systems would be broadly applicable. In its current form, this manuscript now simply describes a reporter prime editing reporter system with some observations about cellular mechanisms needed for prime editing (which as the authors have indicated have been reported by others).

Reviewer #2 (Remarks to the Author):

As previously noted, the authors report a useful tool for optimizing prime editing, which otherwise is a useful but difficult method to use. The data presented support the validation and utility of the optimization tool. Overall, the paper will be useful to people who wish to use prime editing and is thus a valuable contribution. I did raise a number of concerns in my prior review of this study. In their revised manuscript, the authors have addressed my concerns through re-writing, and eliminating less convincing data. I therefore have no additional comments.

Reviewer #1 (Remarks to the Author):

In this revised manuscript, Schene and colleagues address many (but not all) of the critiques of the previous reviews. However, there are some unaddressed concerns that still remain that should be addressed. Also, given that the authors have removed what was considered potentially the most impactful aspect of the manuscript, it is not clear if this manuscript in its current form sufficiently advances the field to warrant publication in Nature Communications.

(1) I still disagree with the authors' assertion that their system should not be compared to reporters of expression (e.g. PE2-SpG-P2A-GFP; Kweon et al Mol Ther. 2021). While it is appreciated that the authors have now made the comparisons between all transfected cells versus fluo PEER-enriched cells, the comparison to cells in which the prime editors are expressed is still a critical one to validate the impact of their reporter system. In other words, enriching cells in which the prime editor is expressed may result in similar levels of editing to that of cells in which the fluo PEER reporter is active. If this were the case, then successful prime editing would not require the cloning and optimization for each site that is required for the fluo PEER system but instead simply enriching for cells in which the prime editor is being actively expressed. Along similar lines, the cellular mechanisms that authors described affecting prime editing might affect expression of the prime editor and not the prime editor's ability to modify genomic loci. However, without this important comparison this cannot be determined.

We thank the reviewer for their critical evaluation of our data. It is considered redundant to compare to cells in which the prime editor is actively expressed (e.g. CMV-PE2-P2A-GFP). This is based on 1) our empirical observations that co-transfection of a mixture of plasmids is >90% efficient and 2) the notion that expression of the fluoPEER cassette and the prime editor are governed by the same cellular mechanisms, since both are transcribed from the same constitutively active CMV promoter. Furthermore, the mechanism suggested by the reviewer has not been described in the literature. Still, we do feel that it is important to clearly distinguish transfection-enrichment (GFP+) from prime editor expression-enrichment (PE2-GFP+). As such, we have added the following caveat to our discussion section: "Given near-complete co-transfection in our experiments (Supplementary Fig. 9), we used a separate GFP-expressing plasmid and not a prime editor plasmid containing a GFP cassette for this comparison."

(2) As stated in the previous review, the authors should include information about indel information and off-target effects in the fluo-peer enriched cells versus the all transfected cells. If enriching for editing is also enriching for indels and off-targets, then this is a severe limitation of the system that the authors do not discuss.

We would like to notify that a comparison of on-target unwanted indels, between fluoPEER-enriched and all transfected cells, had already been included in Figure 3b. This data indicates no increase of unwanted indels using fluoPEER-enrichment. The absence of off-target effects for both groups (all transfected cells and fluoPEER-enriched cells) has now been added as Supplementary Table 1.

(3) It was disappointing to see that the authors removed the shRNA data rather than addressing the previous critiques of the reviewers. This was considered the most impactful component of the manuscript as methods to improve prime editing efficiency in the absence of reporter systems would be broadly applicable. In its current form, this manuscript now simply describes a reporter prime editing reporter system with some observations about cellular mechanisms needed for prime editing (which as the authors have indicated have been reported by others).

We share the notion of reviewer 1 that components that improve prime editing efficiency without the need for reporter systems would be more broadly applicable, for instance for in vivo use. However, we disagree that the investigation of cellular mechanisms influencing prime editing efficiency was the most impactful component of the manuscript. Rather, we agree with reviewer 2 that the most important contribution of the fluoPEER system lies in the easy-to-adapt and rapid optimization and testing of prime editing designs for the generation of specific mutations. Afterwards, the same fluoPEER may be used to enrich for populations of edited cells, facilitating generation of in vitro disease models.

Reviewer #2 (Remarks to the Author):

As previously noted, the authors report a useful tool for optimizing prime editing, which otherwise is a useful but difficult method to use. The data presented support the validation and utility of the optimization tool. Overall, the paper will be useful to people who wish to use prime editing and is thus a valuable contribution. I did raise a number of concerns in my prior review of this study. In their revised manuscript, the authors have addressed my concerns through re-writing, and eliminating less convincing data. I therefore have no additional comments.

We thank the reviewer for their kind words. We agree that the additional validations of the fluoPEER as a prime editing optimization tool has strengthened the core concept of the manuscript. Furthermore, we would like to express our gratitude for their observation that the previous version of the manuscript contained too many unexplained terms, limiting the readability for a broader audience. We feel that re-writing and focusing on the most convincing data have improved the clarity of our manuscript.